# Thiotepa, Busulfan, Cyclophosphamide: Effective but Toxic Conditioning Regimen Prior to Autologous Hematopoietic Stem Cell Transplantation in Central Nervous System Lymphoma

**DOI:** 10.3390/medsci11010014

**Published:** 2023-01-29

**Authors:** Lebon Delphine, Debureaux Pierre-Edouard, Royer Bruno, Gruson Bérengère, Joris Magalie, Votte Patrick, Marolleau Jean-Pierre, Morel Pierre

**Affiliations:** 1Hématologie Clinique et Thérapie Cellulaire, Centre Hospitalier Universitaire Amiens-Picardie, 80000 Amiens, France; 2Hématologie Clinique, Hôpital Saint-Louis, Assistance Publique—Hôpitaux de Paris, 75014 Paris, France

**Keywords:** autologous hematopoietic stem cell transplantation, central nervous system lymphoma, infections, conditioning, adverse events

## Abstract

In primary central nervous system lymphoma, two-year progression-free survival rates of up to 63 percent have been reported for first-line autologous stem cell transplantation after conditioning with the thiotepa busulfan cyclophosphamide regimen. However, 11 percent of the patients died due to toxicity. Besides conventional survival, progression-free survival and treatment related mortality analyses, a competing-risk analysis was applied to our cohort of twenty-four consecutive patients with primary or secondary central nervous system lymphoma who underwent autologous stem cell transplantation after thiotepa busulfan cyclophosphamide conditioning. The two-year overall survival and progression-free survival rates were 78 percent and 65 percent, respectively. The treatment-related mortality rate was 21 percent. The competing risks analysis demonstrate that age 60 or over and the infusion of less than 4.6 × 106/kg CD34+ stem cells were significant adverse prognostic factors for overall survival. Autologous stem cell transplantation with thiotepa busulfan cyclophosphamide conditioning was associated with sustained remission and survival. Nevertheless, the intensive thiotepa busulfan cyclophosphamide conditioning regimen was highly toxic, especially in older patients. Thus, our results suggest that future studies should aim at identifying the subgroup of patients who will really benefit of the procedure and/or to reduce the toxicity of future conditioning regimen.

## 1. Introduction

Since the early 1990s, the standard first-line treatment for primary central nervous system lymphoma (CNSL) has been chemotherapy based on high-dose methotrexate (HD MTX), followed by consolidation with whole-brain radiotherapy; this gives a median overall survival (OS) time of 30 to 50 months [1,2,3,4,5]. The introduction of high-dose chemotherapy with autologous hematopoietic stem cell transplant (ASCT) was based on the combination’s effectiveness in patients with aggressive systemic lymphoma [6]. In patients with CNSL, the use of standard conditioning regimens (such as BEAM: bis-chloroethylnitrosourea, etoposide, cytarabine, melphalan) prior to ASCT has been associated with an unacceptable early relapse rate-possibly because of poor penetration of the central nervous system (CNS) by the drugs used in this combination [7]. Indeed, the blood–brain barrier prevents the diffusion of many cytotoxic drugs into the CNS, apart from certain lipophilic compounds or drugs delivered at very high doses [8,9,10]. Thus, a conditioning regimen based on CNS-penetrating drugs (thiotepa, busulfan, cyclophosphamide) has been evaluated in 43 patients with relapsing, refractory CNSL or in patients who achieved a partial response to first-line treatment. The two-year OS and progression-free survival (PFS) rates were, respectively, 69% and 58% [11]. In a study of 32 patients with primary CNSL, the use of an HD-MTX-based induction regimen and then thiotepa, busulfan, and cyclophosphamide conditioning was associated with an estimated OS rate of 81% after a median follow-up of 45 months [12,13]. However, adverse events (and especially septic complications) were frequent, affecting a third of the treated patients. In patients under the age of 60 with newly diagnosed primary CNSL, the PRECIS trial found a 2-year PFS rate of 63% for the HD-MTX- thiotepa, busulfan, and cyclophosphamide-ASCT combination. However, the treatment-related mortality rate among patients who underwent thiotepa, busulfan, and cyclophosphamide conditioning and ASCT was 13% (n = 5 out of 38) [14]. It should be noted that these results come from prospective clinical trials with highly selected patients. Data in a real-life setting are scarce. Hence, the objective of the present analysis was to evaluate the respective roles of effectiveness and safety of the thiotepa busulfan cyclophosphamide regimen in a real-life cohort of patients with central nervous system lymphoma.

## 2. Patients and Methods

### 2.1. Patient Selection Criteria and Data Collection

All patients aged 18 or over and having undergone thiotepa, busulfan, and cyclophosphamide conditioning/ASCT for primary or secondary CNSL between August 2010 to March 2018 in one of two French university hospitals (in the cities of Amiens and Caen) were included consecutively in the study. The patients’ chemotherapy prescriptions were extracted from the chemotherapy department’s software (CHIMIO^®^, Computer Engineering SARL, Paris, France). The patients’ demographic, clinical, treatment-related and laboratory data were extracted from medical records. Our study complied with the French legislation on retrospective observational studies of clinical practice and with the principles of the Declaration of Helsinki.

### 2.2. The Conditioning and ASCT Protocols

Prior to ASCT on day 0, patients received high-dose chemotherapy with a combination of thiotepa (250 mg/m² per day, from day 9 to day 7), busulfan (3.2 mg/kg per day, from day 6 to day 5, and 1.6 mg/kg on day 4), and cyclophosphamide (60 mg/kg per day, on days 3 and 2). For patients aged 60 and over, the busulfan dose was reduced by 40%. To prevent seizures, clonazepam (0.5 mg/12 h) was administered during busulfan treatment. To prevent hemorrhagic cystitis, mesna was continuously infused during the cyclophosphamide perfusion. Patients received prophylaxis for pneumocystis and viral infections but not for fungal infections. Depending on each center’s practices, some patients received granulocyte colony-stimulating factor during the ASCT.

### 2.3. Patient Follow-Up and Statistical Analysis

Patients were evaluated clinically every day during their hospital stay and then 1 month after discharge, every 3 months for the first year, and every 6 months thereafter. The treatment response was evaluated carefully with magnetic resonance imaging (MRI) 3 months after ASCT and then every 6 months. The primary study’s objective was to evaluate the safety of the thiotepa, busulfan, and cyclophosphamide regimen in a real-life cohort of patients with primary and secondary CNSL, i.e., also in patients who could not be enrolled in the PRECIS trial because they had relapsing primary CNSL or secondary CNSL; both conditions are frequently associated with greater frailty. Since we expected the treatment to be more toxic than in published clinical trials, we chose the treatment-related mortality (TRM) rate as our primary endpoint. Considering the severity of the disease and the TRM rate observed in the PRECIS trial (11%), a TRM rate above 15% or more was considered to be unreasonable. The secondary objective was to evaluate the effectiveness of the thiotepa, busulfan, and cyclophosphamide regimen in this cohort, using the following end points, namely remission rates, OS, and PFS after ASCT. Indeed, OS was defined as the length of time from date of diagnosis and death; PFS was the length of time after treatment without detectable disease.

Post-ASCT OS and PFS were evaluated according to the Kaplan–Meier method, including the 95% confidence interval (95CI). PFS encompassed two potentially competing events, namely progression defined as the progression of lymphoma and TRM, defined as any death occurring before lymphoma progression or relapse. We performed competing risk analyses, taking into account these two events using the cause-specific Cox model and the model proposed by Fine and Gray [15]. For each covariate, the results were expressed as the cause-specific hazard ratio (with a test for statistical significance) for each competing risk. Cumulative incidence plots were fitted using Fine and Gray models [16]. All analyses were performed using SAS software (version 9.4, SAS Institute Inc. Cary, NC, USA, 25513) and R software (release 3.4.1, The R Foundation for Statistical Computing, Vienna, Austria).

## 3. Results

### 3.1. Characteristics of the Study Population

A total of 24 patients were included (Table 1). The median (range) age at the time of ASCT was 58 (23–66). All patients were HIV-negative. Two patients were being treated with MTX for rheumatoid arthritis at the time when the CNSL was diagnosed. A histology assessment revealed diffuse large B cell lymphoma in 22 patients (92%, including 3 transformed follicular lymphoma and 1 transformed small lymphocytic lymphoma) and follicular lymphoma in 2 patients (including 1 with leukemic involvement).

### 3.2. Pre-Transplant Treatment

The treatments received before ASCT are summarized in Table 1. All patients received HD-MTX as first- or second-line chemotherapy, though one of the patients received one or two courses of chemotherapy before ASCT (first-line treatment: *n =* 13, treatment of relapse: *n =* 9; treatment of refractory disease: *n =* 2). All patients received the full-dose thiotepa, busulfan, and cyclophosphamide regimen as described above, except for a 66-year-old patient with secondary CNSL, for whom the busulfan dose was reduced by 40%, as described above.

### 3.3. Engraftment

The median (range) number of infused CD34+ stem cells was 6.07 × 106/kg (0.36–16). Hematopoietic reconstitution was observed in all patients. The median (range) duration of aplasia (defined as a neutrophil count <0.5 G/L) was 14 days (7–37) and the median time to platelet recovery (>20 G/L) was 15 days (9–74). The median (range) number of red blood cell transfusions was 5 (2–16) and the median (range) number of platelet transfusions was 3 (1–10). The median (range) length of hospital stay was 33 days (15–78).

### 3.4. Toxicity of Thiotepa, Busulfan, and Cyclophosphamide/ASCT

Five patients died—all within 3 months of thiotepa, busulfan, and cyclophosphamide conditioning. None had disease progression at the time of death; hence, the TRM rate (95%CI) at 3 months was 21% (9–40). The TRM rate was 6% in patients aged under 60 (1 out of 15) and 45% in patients aged 60 or over (4 out of 9). All five deaths were caused by septic shock; persistent coma was observed in four cases, and acute respiratory distress syndrome was observed in two.

Overall, thiotepa, busulfan, and cyclophosphamide conditioning was associated with a high incidence (100%) of treatment-related adverse events. Sixteen patients (66%) experienced a Common Terminology Criteria for Adverse Events (CTCAE) [17] grade >2 adverse event. These were predominantly infectious and neurological adverse events (Table 2).

All patients presented infections during aplasia (CTCAE grade ≥3 in 41% of the patients), with a high incidence of neutropenic enterocolitis (75%, including 5% with a CTCAE grade 3 event). Fourteen patients (58%) had documented infections, which were recurrent in seven cases, and the remaining had fever of unknown origin. The documented infections included 13 (69%) bacterial infections and—unexpectedly—5 (21%) fungal infections. In three cases, a Candida infection was associated with grade 2 colitis. One patient presented a Candida and Aspergillus co-infection. All the patients with a documented fungal infection died. We also observed two separate cases of cytomegalovirus reactivation and one case of pneumocystis. The infectious events’ CTCAE grade was not correlated with the number of infused CD34+ stem cells. 

Neurological adverse events were observed in 9 patients (37%), including 5 (20%) CTCAE grade ≥3 events. Somnolence was observed in 5 cases, and coma (CTCAE grade 4) was observed in 4 cases (including 1 case of seizures unrelated to busulfan exposure). Neurological adverse events (one case of somnolence and one case of coma) occurred in two of the three patients who had received whole-brain radiotherapy before ASCT.

Nine patients (37%) required intensive care unit admittance (ICU) for coma (*n =* 4), sepsis (*n =* 3), or respiratory failure (*n =* 2). The median length of stay in ICU was 8 days (range, 2–36) with a mortality rate of 55%. 

Other adverse events are also detailed in Table 2. Seven patients had adverse events affecting the skin, of which two were CTCAE grade ≥3. Pharmacological investigations established that thiotepa was the most likely causal agent in 5 of these 7 cases. We observed one case of busulfan-related pulmonary toxicity, but no cases of sinusoidal obstruction syndrome of the liver. 

### 3.5. Outcomes after Thiotepa, Busulfan, and Cyclophosphamide/ASCT

Before thiotepa, busulfan, and cyclophosphamide/ASCT, we observed a complete response (CR) in 15 patients and a partial response (PR) in 9 (Table 1). The median (range) follow-up time was 10 months (0–73). At last follow-up, five patients (21%) had died. The three over-60 patients with a PR before ASCT all died after ASCT. One patient was lost to follow-up 1 year after ASCT and had a CR at the last visit. The 2-year OS and PFS rates were, respectively, 75% and 64% (95CI: 57 to 93% and 44 to 84%, respectively, Figure 1). One patient experienced a CNS relapse 3 months after ASCT; whole-brain radiotherapy then gave a persistent CR. Another patient experienced a CNS relapse 14 months after ASCT, and treatment with rituximab and bendamustine led to a persistent CR at the last follow-up (4 months later). No systemic relapses were observed. At the end of the study, the 19 living patients had a CR. 

### 3.6. Prognostic Factors Associated with Survival

Age 60 or over was the only statistically significant adverse prognostic factor for PFS (*p* = 0.049, Figure 2). The number of courses of chemotherapy before ASCT, PS before ASCT, and the number of CD34+ stem cells infused did not have significant prognostic value. Age of 60 or over and the infusion of less than 4.6 × 106/kg CD34+ stem cells (median of CD34+ stem cells infused) were significant adverse prognostic factors for OS (*p* = 0.05). The number of courses of chemotherapy before ASCT and PS before ASCT did not have significant prognostic value.

Cause-specific Cox models with TRM and relapse as competing events indicated that age 60 or over, infusion of less than 4.6 × 106/kg CD34+ stem cells, and the absence of a CR before ASCT were borderline significant adverse prognostic factor for TRM (Table 3), whereas no prognostic factors were found for the risk of disease progression—probably because of the very small number of events. The cumulative incidences of progression and progression-free death by age are shown in Figure 3.

## 4. Discussion

As expected, the 21% 3-month TRM rate reported in our real-life setting was higher than that reported in the PRECIS clinical trial [14,18]. Moreover, our TRM is also higher than that usually reported with BEAM conditioning (0–10%) [19]. 

Overall, thiotepa, busulfan, and cyclophosphamide/ASCT leads to serious complications, as shown by the high incidence (66%) of CTCAE grade ≥3 adverse events. Most of these events were infections (41%) and neurological adverse events (20%), and they occurred more frequently in older patients. 

In similar studies, the TRM and CTCAE grade 3/4 infection rates ranged, respectively, from 0 to 11% [12,20,21,22] and 20 to 27% [11,20,21], depending on the study population. We observed that 21% of the patients had a documented fungal infection; this proportion was unexpectedly high, relative to the rate of 1% typically reported after ASCT [23]. The most common fungal infection was invasive candidiasis, which might be related to the prolonged neutropenia and colitis. The frequent use of high-dose corticosteroids during the pre-ASCT chemotherapy might also increase the risk of infection—especially in patients who have already been treated intensively. Antifungal prophylaxis with fluconazole might decrease the risk of invasive candidiasis before neutrophil recovery in TBC/ASCT and we used it systematically after this study. However, as specified in the recent update of the recommendations of the Infectious Diseases Working Party (AGIHO) of the German Society of Hematology and Medical Oncology (DGHO), the available data do not support the prophylactic use of antifungals to prevent invasive fungal disease (IFD) in ASCT. ECIL conferences also do not recommend for patients undergoing autologous HSCT, an antifungal prophylaxis, for whatever underlying condition, as these patients are at low risk of IFD [24]. However, fluconazole should be considered to prevent mucosal Candida infection during the neutropenic phase (grade B-III). IFDs are rare events after HDC/ASCT and no reduction in mortality has been found in patients after HDC/ASCT [25]. 

In some published studies, neurological adverse events were not observed [22,26]. In contrast, other studies reported encephalopathy in 11% to 18% of patients (mostly those aged over 60 or having undergone radiotherapy) [20] and a case of grade 3 reversible encephalopathy [12]. In our series, neurological adverse events were frequent: coma occurred in 45% of the patients with neurological complications. This high incidence might be related to the age and perhaps frailty of our study population (37% were over the age of 60) and the high proportion of patients having already received whole-brain radiotherapy. Once again, the frequent use of high-dose corticosteroid during the management of CNSL might also have a role in the onset of neurological adverse events in these patients. The CNS penetration of thiotepa and busulfan might also be associated with neurological adverse events in vulnerable patients (i.e., those aged 60 or over or having received whole-brain radiotherapy). Given that busulfan and thiotepa are known to penetrate well into the CNS (cerebrospinal fluid levels are at least 80% of the serum levels for both drugs [27]), thiotepa, busulfan, and cyclophosphamide conditioning was highly effective in our real-life cohort: the 2-year PFS rate after thiotepa, busulfan, and cyclophosphamide/ASCT was 65%—even when considering patients with secondary CNSL or relapsing primary CNSL. 

Finally, our competing risk study identified two adverse prognosis factors for survival: age of 60 or over and the infusion of less than 4.6 × 106/kg CD34+ stem cells, and very interestingly, these factors are the only involved on TRM. That suggested our patients died of toxicity and not of disease and then could leads us to better select patients for this type of conditioning. It is important to underlie that in the PRECIS trial the median age was 55 (25–60) in the ASCT arm. A recently presented study highlights the importance of ASCT in the management of CNSL with a clear survival benefit compared to chemotherapy alone [10,28,29,30] or whole blood radiotherapy [18]. These data confirmed the importance of ASCT no matter age and encourage us to find the best conditioning regimen with less toxicity. Ongoing trial evaluate the risk/benefit ratio of ASCT conditioning by thiotepa-busulfan without cyclophosphamide (ClinicalTrials.gov identifier: NCT04446962).

## 5. Conclusions

Although the incidence of treatment-related adverse events was greater than in clinical trials, our present results suggest ASCT with a thiotepa, busulfan, and cyclophosphamide-based conditioning regimen can be used routinely to treat secondary CNSL or relapsed primary CNSL, as has already been reported for first-line treatment of primary CNSL. Infectious complications were frequent, and the high incidence of invasive candidiasis argues in favor of routine antifungal prophylaxis. Nevertheless, this study shows that older age significantly impacts OS and TRM, suggesting patients died due to toxicity. Use of a less intensive conditioning regimen should now be evaluated in this patient population to reduce TRM without compromising the effectiveness. Thus, our results suggest that future studies should aim at identifying the subgroup of patients who will really benefit of the procedure and/or to reduce the toxicity of future conditioning regimens.

## Figures and Tables

**Figure 1 medsci-11-00014-f001:**
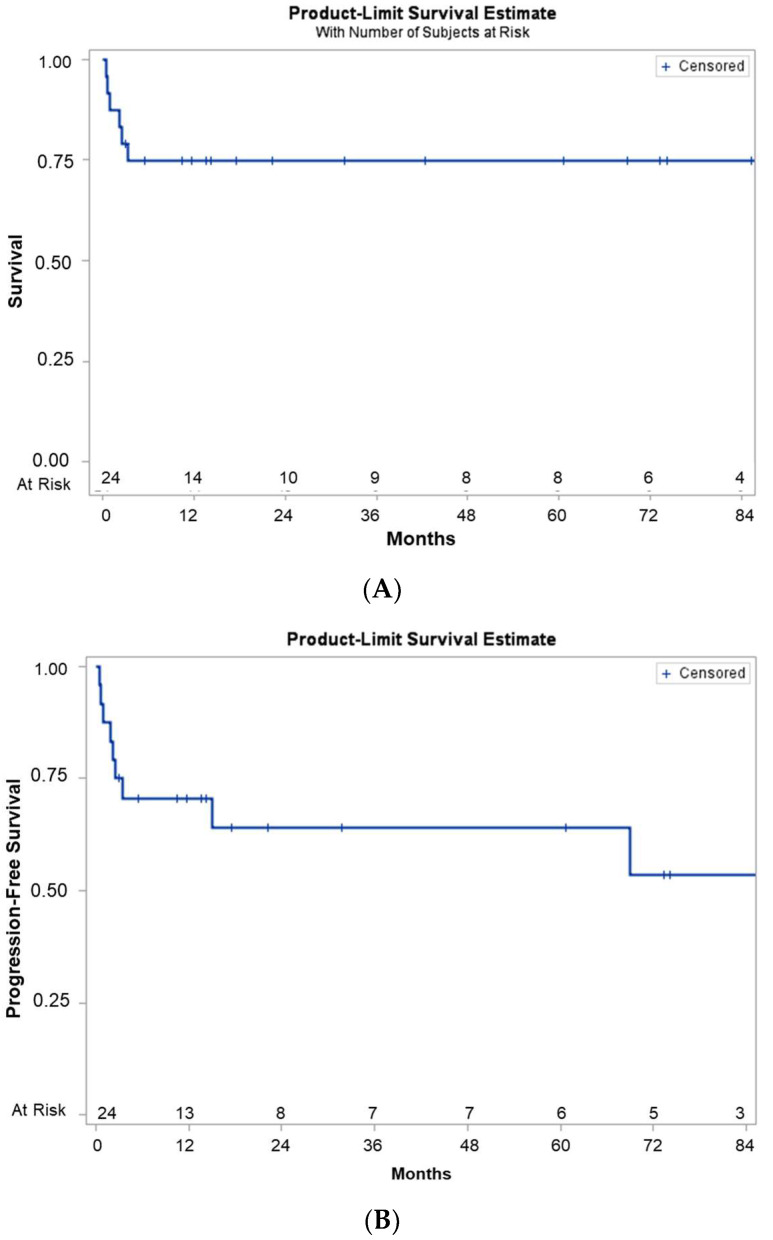
The OS rate (**A**) and the PFS rate (**B**).

**Figure 2 medsci-11-00014-f002:**
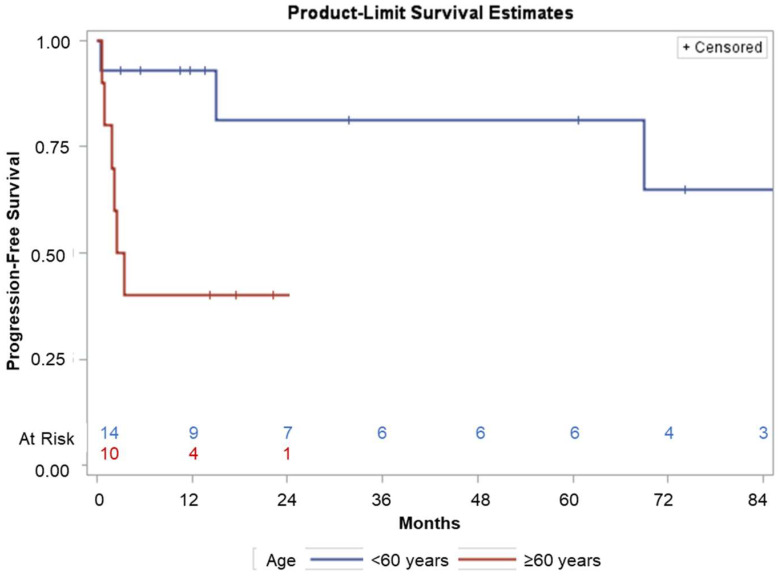
The PFS curves, by age.

**Figure 3 medsci-11-00014-f003:**
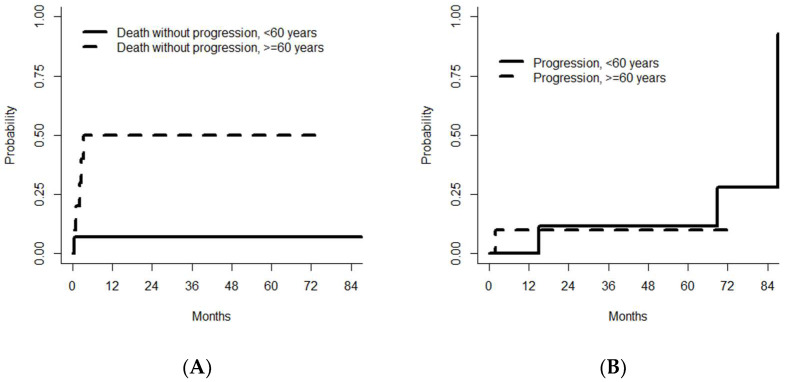
Cumulative incidence of death without progression (i.e., TRM) (**A**) and disease progression (**B**), by age.

**Table 1 medsci-11-00014-t001:** Patient characteristics on inclusion.

	Primary CNSL (*n* = 15)	Secondary CNSL (*n =* 9)	Whole Study Population (*n* = 24)
Age at diagnosis, years, median (range)	55	57	56.5
Sex (%)			
Male	10 (66)	5 (56)	15 (63)
Female	5 (33)	4 (44)	9 (38)
PS at inclusion (%)			
0	3 (20)	0 (0)	3 (13)
1–2	7 (47)	4 (44)	11 (46)
3–4	1 (7)	1 (12)	2 (8)
Missing data	4 (26)	4 (44)	8 (33)
Ocular involvement (%)	4 (26)	1 (11)	5 (21)
CSF positive (%)	1 (6)	3 (33)	4 (17)
Histology (%)			
Diffuse large B-cell lymphoma	14 (93)	8 (89)	22 (92)
Follicular lymphoma	1 (6)	1 (11)	2 (8)
Number of courses of chemotherapy (%)			
1	11 (73)	3 (33)	14 (58)
2	4 (26)	5 (56)	9 (38)
>2	0 (0)	1 (11)	1 (5)
Intrathecal treatment (%)	0 (0)	4 (44)	4 (17)
Radiotherapy (%)	2 (13)	1 (11)	3 (5)
Indication for ASCT (%)			
First-line treatment	10 (66)	3 (33)	13 (54)
Relapsed disease	4 (26)	5 (56)	9 (38)
Refractory disease	1 (6)	1 (11)	2 (8)
Time interval between diagnosis and ASCT, months, median (range)	6 (3–100)	20 (5–81)	6 (3–100)
Age at the time of ASCT (%)			
≤60 60–65	9 (60) 3 (20)	5 (55) 3 (34)	14 (58) 6 (25)
>65	3 (20)	1 (11)	4 (17)
Response status at the time of ASCT (%)			
CR	9 (60)	6 (67)	15 (63)
PR	6 (40)	3 (33)	9 (38)
PS at the time of ASCT * (%)			
0	9 (60)	5 (56)	14 (58)
1	6 (40)	3 (33)	9 (38)

ASCT: autologous stem cell transplantation, PS: performance status, CR: complete response, PR: partial response. *, missing data.

**Table 2 medsci-11-00014-t002:** Adverse events, according to the CTCAE criteria.

	Grade 1/2	Grade 3	Grade 4	All
Infections	13	2	9	24
Neurological events	4	1	4	9
Mucositis *	5	7	3	15
Cutaneous events	5	2	0	7
Other complications				
Colitis	18 out of 24 (75%) had a CTCAE grade 3 event			
Renal dysfunction	2 out of 24			
Hemorrhagic cystitis	2 out of 24			

* Missing data.

**Table 3 medsci-11-00014-t003:** Competing risks analysis.

		Fine and Gray Model					Cause Specific Hazard Model	
Covariate	Reference	3-yr CIF	Hazard Ratio	LCL	UCL	*p* Value	Hazard Ratio	*p* Value
Age			8.003	0.89	72.9	0.06	8.72	0.048
<60	reference	0.07		0.06	0.08			
>=60		0.5		0.44	0.56			
cd34			0.12	0.014	1.12	0.06	0.13	0.05
<4.6	reference	0.5		0.44	0.56			
>=4.6		0.07		0.06	0.08			
status before ASCT			4.57	0.96	21.8	0.05	4.41	0.08
CR	reference	0.13		0.11	0.15			
PR		0.44		0.38	0.5			

Abbreviations: CIF, Cumulative incidence function; LCL, Lower confidence limit; UCL, Upper confidence limit; ASCT, Autologous stem cell transplantation; CR: complete response, PR: partial response.

## Data Availability

Not applicable.

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
