# Peer review of "Thiotepa, Busulfan, Cyclophosphamide: Effective but Toxic Conditioning Regimen Prior to Autologous Hematopoietic Stem Cell Transplantation in Central Nervous System Lymphoma"

_medsci, 2023, doi:10.3390/medsci11010014_

Round 1
Reviewer 1 Report
In this study, Dr. Delphine and colleagues reported outcomes of 24 patients with primary and secondary CNS lymphoma receiving autologous stem cell transplantation using thiotepa, busulfan and cyclophosphamide (TBC) regimen.
While this regimen was not associated with prolonged BM suppression with median time to neutrophil engraftment of 14 days, high rate of severe toxicities and TRM were observed with 21% TRM at 3 months, leading to the conclusion that this regimen is highly toxic. However, in my opinion, majority of toxicities reported in this study could have been prevented such as skin toxicity from thiotepa or bacterial and fungal infections during neutropenic period.
- In the manuscript, the authors stated that “Patients received prophylaxis for pneumocystis and viral infections but not for fungal infection. What about antibacterial prophylaxis? Was antifungal prophylaxis commenced in patients who experience mucositis? Even though the role of antifungal prophylaxis in autologous stem cell transplantation is controversy, several studies and practice guidelines (such as NCCN) recommend fluconazole or echinocandin prophylaxis in patients with mucositis. The authors should add this in the discussion. This could explain the high incidence of fungal infections in this study.
- The incidence of skin toxicity from thiotepa was quite high in this study compares with the other studies using the same intensity of thiotepa.
Are there preventive measures implemented in the SOP?
Other comments
- In the statistical method, the authors stated that cause specific Cox model was used, and the results were expressed as cause-specific hazard ratio but I don’t see cause-specific HR presented (and 95%CI) in the result sections. Only P values were reported.
- When cumulative incidence with competing risk analysis was used, TRM and relapse should be presented as a cumulative incidence with 95%CI at given time points.
- To avoid confusion, please clarify the phase…. “To assess the respective impacts of disease progression and TRM on OS and the impact of severe toxicity (defined as admission to an intensive care movement) on TRM, we performed an original unpublished competing risks analysis.”
Did the authors use “disease progression and TRM” as independent variables in the regression model for OS (dependent variable)?
Similary, was “severe toxicity” used as an independent variable in the regression model for TRM?
- The description of figure 3A should be “cumulative incidence of TRM”, not death which means “all-cause mortality”
- Given high TRM rate reported, comorbidities or HCT-CI should be reported in Table 1.
- The transplant period was from 2010-2018. Is there any explanation for a very short median follow up duration (10 months)?
- The long-term follow up results of the PRECIS study have been recently reported. The authors should update the reference list. (DOI: 10.1200/JCO.22.00491 Journal of Clinical Oncology 40, no. 32 (November 10, 2022) 3692-3698.)
Reviewer 2 Report
In this article, the Authors described the efficacy of TBC, an intensive conditioning regimen prior to ASCT, in adult patients with CNSL. The subject of the study is relevant, since there is no extensive literature on this rare lymphoma. However, a revision of the paper is required, considering the following issues:
- the characteristics of patients should be better described, in particular the treatments before ASCT (only HD-MTX is mentioned);
- the authors should also provide more details on the reported infections. The 100% infection rate is too high, even for this particularly fragile and high-risk population. Authors should specify whether FUO was observed
- more details on statistical methods are required, in particular what are the competing risks considered in the analysis; it is assumed: relapse-related vs non-relapse related death; the Authors should also declare all parameteres considered in the competing risk analysis
- in the Results section for any significant prognostic factors the HR and 95%CI must be reported; a dedicated table could be useful
- line 41: references [11-13] are not appropriated and are unrelated to the text (which only describes results of a study with 43 patients),
- line 46: the proper reference to the PRECIS trial is missing
- line 76: MRI has not been defined
- Table 1: layout needs to be improved. In the line "Radiotherapy" there is a mistake in the sum reported in "whole study population", it must be 3.
- line 114: /Kg must be added after the infused CD34.
Round 2
Reviewer 1 Report
There are some issues need further clarification.
1. Table 1
-Number of missing data should be reported for example performance status.
-Also percentages of PS are missing.
-Ranges of time interval from diagnosis and transplant and age at diagnosis are missing.
-What is the difference of PS at inclusion and PS at transplant? Weren't all patients included at the time of transplant?
-As per my previous comment, the HCT-CI was not added in the table 1. Only PS was added.
2. The new added table "Supplemental table. Competing risks analysis". -Please specify the outcome for this analysis; for relapse or TRM? Also, if this table is added in the main text, it should not be "supplemental table".
-What are RC, RP stated in this table?
3. All KM curves should be presented with number of patients at risk at each time point.
4. The authors should discuss on the recommendations on antifungal prophylaxis in the AGIHO guidelines cited in this paper. In the guidelines, only 2 references were cited and none of these references compared antifungal prophylaxis vs. no prophylaxis in "autologous stem cell transplant" setting especially in the setting of auto-SCT with high risk of mucositis.
5. Why did the authors used the cutoff of CD34 at 4.6 in the analysis?
6. Please discuss on the prevention of skin toxicity of thiotepa. As in my previous comment, the skin toxicity was quite high, and might be preventable by using proper skin cares during the infusion.
Reviewer 2 Report
I have read the revised version of the manuscript. Since the Authors responded fully to all comments, I confirm that the manuscript can be considered for publication in the revised form.
Author Response
Thank you for the reviewing and this positive decision.